# Microbial diversity drives carbon use efficiency in a model soil

Luiz A. Domeignoz-Horta [1✉], Grace Pold [2], Xiao-Jun Allen Liu [1], Serita D. Frey[3], Jerry M. Melillo[4] & Kristen M. DeAngelis [1✉]

Empirical evidence for the response of soil carbon cycling to the combined effects of warming, drought and diversity loss is scarce. Microbial carbon use efficiency (CUE) plays a central role in regulating the flow of carbon through soil, yet how biotic and abiotic factors interact to drive it remains unclear. Here, we combine distinct community inocula (a biotic factor) with different temperature and moisture conditions (abiotic factors) to manipulate microbial diversity and community structure within a model soil. While community composition and diversity are the strongest predictors of CUE, abiotic factors modulated the relationship between diversity and CUE, with CUE being positively correlated with bacterial diversity only under high moisture. Altogether these results indicate that the diversity × ecosystem-function relationship can be impaired under non-favorable conditions in soils, and that to understand changes in soil C cycling we need to account for the multiple facets of global changes.

[1] Department of Microbiology, University of Massachusetts, Amherst, MA 01003, USA. [2] Graduate Program in Organismic and Evolutionary Biology, University of Massachusetts, Amherst, MA 01003, USA. [3] School of Natural Resources and the Environment, University of New Hampshire, Durham, NH 03824, USA. [4] The Ecosystems Center, Marine Biological Laboratories, Woods Hole, MA 02543, USA. ✉email: ldomeignozho@umass.edu; deangelis@microbio.umass.edu

The provision of ecosystem functions is dually threatened by human-induced climate change[1–3] and biodiversity loss[4,5]. One such function threatened by these factors is the storage of organic carbon (C) in soils[6–9], which is crucial for climate regulation[3]. This C stock is regulated in part by the rate and efficiency with which the microbes living within soil incorporate fresh plant inputs into their biomass and more stable components of soil organic matter[10,11]. Indeed, predictions of soil carbon stocks under warming are highly sensitive to the assumptions made about microbial carbon use efficiency (CUE)[7,12–14], which is the fraction of C taken up by microbial cells and retained in biomass as opposed to being respired. CUE can be directly affected by global changes such as climate warming and shifts in soil moisture due to modifications in precipitation regimes[3,15]. Meanwhile, global changes are also driving shifts in the diversity and structure of microbial communities[16,17]. Understanding the drivers of CUE is crucial to determine the fate of C in the soil. However, it is uncertain how these direct and indirect impacts of global changes are driving CUE in soils. Factors such as temperature, moisture, microbial community structure, substrate quality, substrate availability, and soil physico-chemical properties are all likely to affect CUE[11,15,18–20], but parsing out their relative importance in natural ecosystems remains a challenge[21].

Climate change is impacting soil temperature and water availability, which are known to directly influence microbial metabolism and can therefore impact CUE. Generally, elevated temperatures increase respiration more than growth, and therefore CUE tends to decrease with increasing temperature[18]. However, this decrease in CUE with temperature is not ubiquitous[22], and has been observed to vary with substrate quality[20]. Our knowledge of the impact of soil water content specifically on CUE is limited to two studies[19,23]. Normally, soil microbial communities living in drier soils are expected to have higher metabolic costs due to osmoregulatory mechanisms[24] such as production of intracellular solutes[24]. Another response to drought is the production of extracellular polysaccharide (EPS), which might also imply in further costs[15]. In addition, low water availability can decrease substrate supply to microbial cells due to slow diffusion rates[25] resulting in a greater proportion of substrate allocated for maintenance metabolism and less available to growth. In either case, moisture limitation is expected to reduce CUE[15]. In addition to these direct effects of abiotic factors on CUE, temperature and moisture can also drive changes in microbial diversity[17] and community structure[26], thus indirectly impacting CUE.

The impact of diversity and community structure for microbial CUE remains unexplored. Positive relationships between diversity and soil functions have been observed, for example, for denitrification[27–29] and methanogenesis[30,31], which are soil functions attributed to relatively restricted groups of microorganisms. Broader soil processes, such as C cycling, are considered to show extensive functional redundancy and be less subject to changes in diversity[32,33]. However, some studies have demonstrated that even broad processes within C cycling can show a positive relationship with diversity, as has been shown for respiration[8,34] and decomposition[35]. Moreover, community composition, rather than richness, can have a large impact on C cycling in soils[32,36]. CUE is known to differ between bacterial strains grown under identical conditions[22]. This suggests that communities with distinct members could have different community CUE. Moreover, it has been shown that abiotic factors (e.g., temperature) can modulate the relationship between diversity and growth in liquid cultures[2], but it remains unclear how temperature and moisture could alter the relationship between diversity and growth efficiency in soils. In the context of global change, it is crucial to better understand how CUE is subject to changes in microbial diversity and community composition.

The overall aim of this study is to provide empirical evidence for the response of CUE to the combined effects of temperature, moisture, diversity loss and distinct community compositions. We hypothesize that: (1) more diverse soil communities have higher CUE compared to less diverse soil communities; (2) an increase in temperature or decrease in soil moisture both reduce CUE and (3) abiotic conditions modulate the relationship between diversity and CUE. To overcome the challenges in determining the response of CUE to environmental factors that co-vary across space and time in natural soils, we develop a model soil (described in "Methods") to control and manipulate the desired variables. We extract microbial communities from field soil collected from a temperate deciduous forest at the Harvard Forest Long-term Ecological Research (LTER) site. We manipulate the diversity of the extracted microbial community in one of three ways prior to inoculation: (1) diversity removal approach[37] with three diversity levels (non-diluted "D0," 1000× diluted "D1," and 100,000× diluted "D2"); (2) filter to 0.8 µm to exclude fungi and have predominantly bacteria ("bacteria only treatment," or "B$_{only}$"); and (3) enrichment for spore-forming organisms[38] ("SF"). These communities are inoculated into the model soil and incubated for 120 days under two different temperatures (15 and 25 °C), and two soil moistures (30 and 60% water holding capacity (WHC)), in a full factorial design totaling 200 samples (Fig. 1). At the end of the incubation, we measure CUE using the $^{18}$O–H$_2$O method[39] and assess bacterial and fungal diversity. We also measure three additional parameters that are proposed to affect CUE. Potential activity of the extracellular enzymatic pool is measured as a proxy for enzyme production[40]. The ratio of ITS to 16S rRNA is used to estimate the fungal:bacterial ratio[41]. Soil aggregation is measured as a proxy for substrate supply. For example, under low water content connectivity is greater within than between aggregates while under higher water content connectivity is increased more between aggregates than within an aggregate[42]. We find that bacterial phylogenetic diversity is positively correlated with CUE under high but not low soil moisture. Using path analysis to distinguish between direct and indirect drivers of CUE, we find that temperature and moisture indirectly influence CUE by altering microbial community structure, but it is the microbial components that directly explain CUE. Our work shows that the impact of diversity on CUE depends on soil moisture, indicating a dynamic interplay between the abiotic and biotic drivers of CUE.

## Results and discussion

**Microbial community assembly in model soils.** Representatives of four bacterial and three fungal phyla grew in the model soil, with 1036 bacterial operational taxonomic units (OTUs) (100% identity), and 270 fungal OTUs (97% identity). The experimental manipulations successfully altered microbial diversity, with higher bacterial and fungal richness in the communities derived from the least diluted inocula (D0) compared to all the others (Fig. 1 and Supplementary Figs. 1 and 2). However, as previously observed[27], the decrease in diversity was not commensurate with the degree of dilution: similar reductions in diversity were observed for dilutions of three (D1) and five (D2) orders of magnitude for both bacteria 77.4% (CI$_{95\%}$ = [63.4–90.6%]) vs. 80.9% (CI$_{95\%}$ = [71.6–106.2%]), and fungi 57.9% CI$_{95\%}$ = [45.4–70.5%]) vs. 43.9% (CI$_{95\%}$ = [19.5–68.3%]). Filtering the D0 inoculum at 0.8 µm was overall successful at removing eukaryotic cells ("B$_{only}$"), as most samples showed zero fungal richness in this treatment (Fig. 1 and Supplementary Fig. 2). The presence of fungal sequences in some B$_{only}$ samples could suggest that some

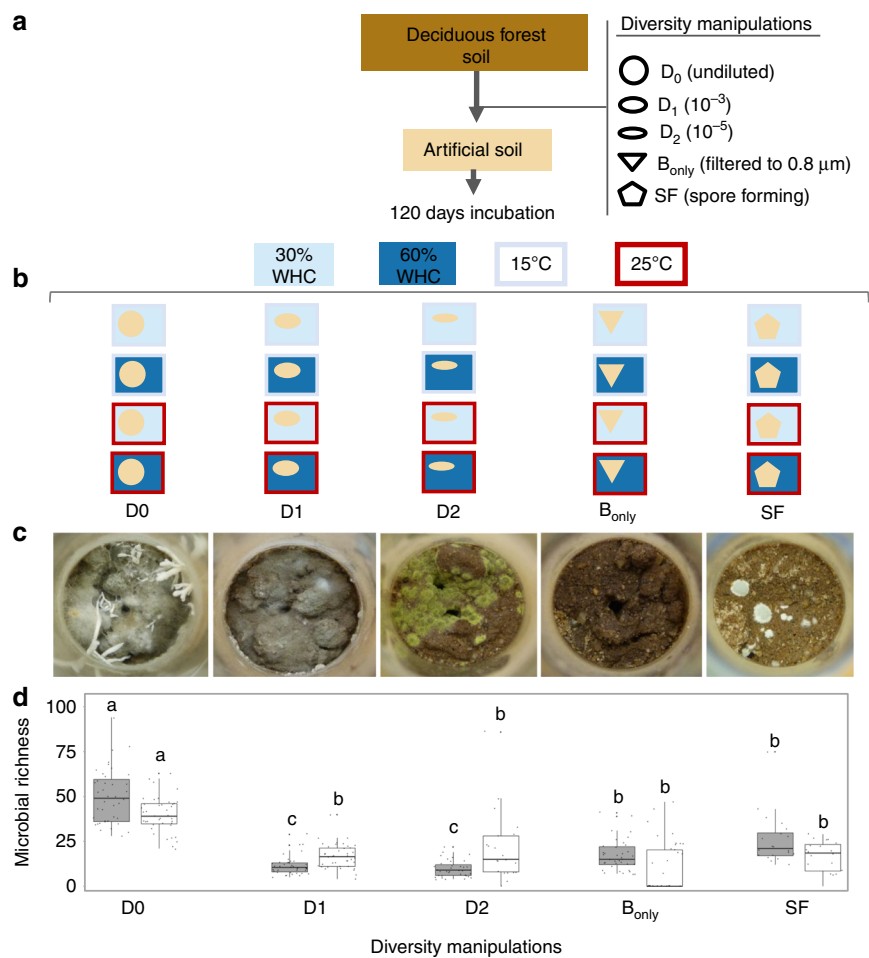

**Fig. 1 Experimental design for manipulation of microbial diversity.** The microbial diversity of a soil inoculum obtained from a temperate deciduous forest was manipulated by (1) sequential dilutions; (2) excluding fungi ("$B_{only}$"); and (3) selecting for spore-forming microorganisms (SF) (**a**). These inocula were added to artificial soil incubated for 120 days under two moisture (30 and 60% water holding capacity) and two temperature (15 and 25 °C) regimes (**b**). Images of model soils at the end of incubation (**c**). Average bacterial (gray) and fungal (white) richness (operational taxonomic units) for each diversity treatment (**d**). Significant differences between treatments within a microbial group (bacteria or fungi) are indicated by different letters (one-way ANOVA followed by Tukey HSD test, $P < 0.05$, df = 171, $n = 176$ for bacteria and for fungi df = 156, $n = 161$). In the boxplots, whiskers denote the minimum value or 1.5× interquartile range (whichever is more extreme), and box denotes interquartile range. The horizontal line denotes the median. Points indicate biological replicates, $n = 40$ and 40 for D0 and D1, 35 and 21 for D2, 38 and 40 for $B_{only}$ and 23 and 20 for SF for bacteria and fungi, respectively.

spores present in the initial soil resisted sterilization, although none of our uninoculated controls indicated growth (described in "Methods"). Finally, we successfully enriched communities in spore-formers (SF) by subjecting the same amount of inoculum soil as was used in D0 to dry heat (120 °C for 30 min) and phenol (1.5% for 1 h), as evidenced by the significantly higher relative abundance of Firmicutes in relation to all other treatments ($F = 338.3$, df = 172, $P < 0.0001$). This vigorous pretreatment reduced the size of the inoculum to such a degree that no growth was observed in the low moisture treatment at 15 °C, and only four replicates showed growth at the high moisture treatment at 15 °C (Supplementary Figs. 1 and 2).

We evaluated fungal and bacterial abundance at the end of the 120 day incubation by real-time quantitative PCR (qPCR) of a bacterial and a fungal house keeping gene. Bacterial and fungal gene copy number did not statistically differ between treatments except for the $B_{only}$ treatment which showed a significantly higher number of bacterial cells and lower fungal numbers than all other treatments (Supplementary Figs. 3 and 4, respectively).

We were able to generate communities with distinct diversity and community structure within the model soil. Richness was predominantly driven by the diversity manipulations, while community structure was responsive to soil moisture and temperature manipulations (Table 1 and Supplementary Figs. 5–8). The microbial richness in the model soil was lower than in natural soils[43], but greater than previous studies aiming to evaluate the relationship between diversity and ecosystem processes[2,8,32]. While caution is needed when interpreting our findings in relation to natural soils, by using a soil-mimicking matrix we were able to begin to address how biotic and abiotic factors interact to drive microbial CUE in a spatially structured soil environment[2,8].

**Empirical link between diversity and CUE.** We measured community CUE with the substrate independent $^{18}O–H_2O$ method[39] under the same temperature and moisture conditions the samples had been incubated at for the previous four months. CUE varied across the range of values measured in other studies[18] (Supplementary Fig. 9). We hypothesized that CUE is positively correlated with diversity. Overall, we observed higher community CUE in the most diverse treatment (D0) compared with communities derived from the first (D1) and second dilutions (D2) (Supplementary Fig. 9), and lower CUE in $B_{only}$ compared to all other treatments.

**Table 1 Percentage of variance explained by the diversity manipulations (Div), moisture (Mois) and temperature (Temp) treatments, and their interactions for bacterial and fungal alpha diversity metrics and bacterial and fungal community structure.**

| Parameter | Div | Mois | Temp | Div: Mois | Div: Temp | Mois: Temp | Div:Mois: Temp | Residuals |
|---|---|---|---|---|---|---|---|---|
| Bacterial diversity (PD) | 49.81*** (0.001) | 0.04 (0.714) | 0.39 (0.266) | 1.73 (0.239) | 2.46· (0.093) | 1.61* (0.025) | 1.88 (0.117) | 42.08 N/A |
| Fungal diversity (Shannon) | 33.37*** (0.001) | 0.11 (0.596) | 0.05 (0.735) | 2.68 (0.192) | 2.27 (0.26) | 1.12 (0.116) | 0.95 (0.54) | 59.44 N/A |
| Bacterial community structure | 29.74*** (0.001) | 10.57*** (0.001) | 2.80*** (0.001) | 5.96*** (0.001) | 2.78* (0.036) | 0.53 (0.163) | 1.41 (0.166) | 46.22 N/A |
| Fungal community structure | 21.81*** (0.001) | 2.92*** (0.001) | 1.52* (0.015) | 1.91 (0.503) | 1.86 (0.49) | 0.24 (0.863) | 1.11 (0.741) | 68.62 N/A |

Bacterial and fungal community structures correspond to the first axis of a non-metric multidimensional scaling analysis (NMDS). The percentage of explained variance is obtained by dividing the group sum of squares by the total. Significant variables are indicated (·$P < 0.01$, *$P < 0.05$, ***$P < 0.0001$), and the exact $P$ values are shown below each explained variance.

CUE represents the allocation of C to growth versus respiration, and to understand how it is affected by diversity, we separately evaluated growth and respiration responses. We observed no significant relationship between fungal diversity and CUE. Regarding bacteria, under high moisture conditions, growth rate increased faster with phylogenetic diversity (PD) (Fig. 2a) than did respiration (Fig. 2b), leading to a significant positive relationship between bacterial phylogenetic diversity and CUE (Fig. 2c) in moist but not dry soils. Interestingly, CUE appeared to be constrained to high values in soils with high bacterial diversity (50–80%). This was confirmed by a break point analysis which showed a threshold at a PD value of 4.48 after which only high CUE values were observed ($t = 4.51$, df $= 86$, $R^2 = 0.28$, $P < 0.001$). By contrast, the lower diversity samples showed the full range of CUE values suggesting that other factors such as community composition[22] and environmental factors are important in determining community CUE. While we report a positive relationship between diversity and CUE, other studies have evaluated the relationship between respiration and diversity, and found it to be positive[8,44], neutral[44,45], or negative[46]. Fewer studies have evaluated the relationship between diversity and growth rate and/or CUE[11,47]. A previous study found no relationship between microbial community composition based on phospholipid fatty acid (PLFA) analysis and CUE[47], though PLFA has much lower resolution compared to sequencing for measuring community composition. The disparities in responses between diversity × C-cycle functions in different studies may be due to the distinct levels of diversity within different experiments as suggested in a recent review[32], where it was concluded that a positive relationship between diversity and C-cycle functions is only consistently observed for low diversity communities (<10 species)[32]. However, we only observed no significant relationship between diversity and CUE after a PD value of 4.48. This indicates that in our simplified soil-mimicking system a moderate level of diversity was required to ensure high CUE. While the level of diversity observed in natural soils is higher than in our simplified system, the complexity of natural soils is also higher. For example, microorganisms living in natural soils experience a multitude of different substrates while in our simplified system we used a single substrate (cellobiose). This suggests that natural soils may require higher levels of diversity before the relationship between diversity-CUE saturates. Overall, these results suggest that by evaluating the diversity–function relationship in a soil-mimicking system, the level of diversity needed to saturate this relationship was higher compared to less complex environments such as found in liquid cultures[2,32]. Moreover, because we used a solid-matrix system, we were able to evaluate how the interplay between biotic and abiotic conditions shapes this component of C cycling.

A complementarity effect may explain why we observed a positive relationship between diversity and CUE under moist and

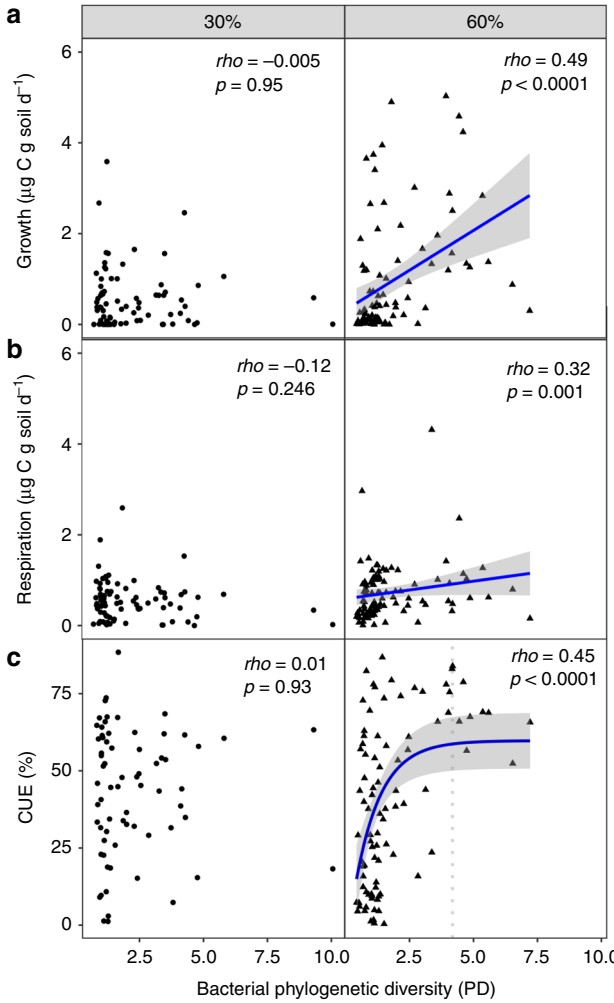

**Fig. 2 Relationship between bacterial diversity and growth, respiration and CUE.** Relationship between bacterial phylogenetic diversity (PD) and growth (**a**), respiration (**b**), and CUE (**c**). Microcosms incubated under 30 and 60% WHC are shown on the left and right panels, respectively. Monotonic relationships between the diversity metric and growth, respiration or CUE are evaluated with Spearman correlation and when significant are indicated with a blue line. We fit linear curves for growth and respiration. Biologically, CUE cannot be >100%, thus we fit a saturating curve to the CUE data. The vertical dashed line indicates the threshold at which there is no more significant relationship between bacterial diversity (PD) and CUE. Shaded area denotes 95% confidence intervals. There were 84 and 92 replicates for 30% and 60% WHC, respectively.

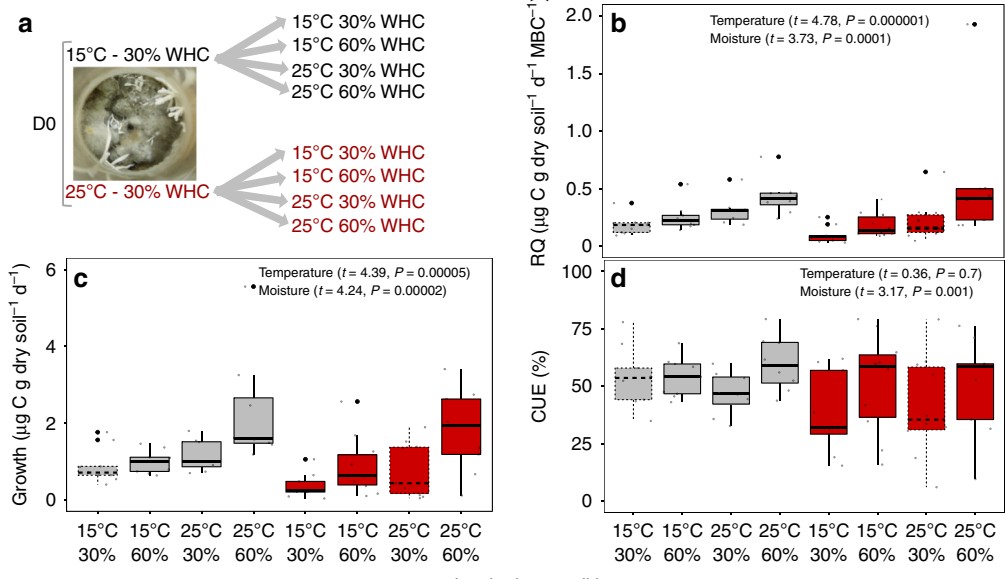

**Fig. 3 Effect of short-term changes in temperature and moisture on respiration, growth and CUE.** Microbial communities from the less diluted treatment (D0) grown at both temperatures (15 and 25 °C) and at 30% water holding capacity (WHC) were incubated under all combinations of water content and temperatures (experimental outline; **a**). Influence of moisture and temperature shifts on respiratory quotient (RQ; **b**), growth (**c**), and CUE (**d**) in the model soils. We used linear mixed effect models to evaluate the impact of short-term changes in abiotic conditions on respiration, growth and CUE with microcosm as the random effect ($n = 72$, df $= 51$). Dashed boxplots represent the long-term soil incubation conditions. In the boxplots, whiskers denote the minimum value or 1.5× interquartile range (whichever is more extreme), and box denotes interquartile range. The horizontal line denotes the median. Points represent individual biological samples ($n = 8$ for each incubation condition).

not dry conditions. Complementarity effects arise from facilitation and niche differentiation that resulted from inter-species interactions increasing overall community productivity[48]. The influence of complementarity effect on function has been previously shown to vary with abiotic conditions[2]. In our study, we propose that the aqueous phase acted as a "gatekeeper" of microbial interactions[49,50] allowing species interactions and a complementary effect to emerge in the high moisture but not low moisture treatment. A mechanism that could explain complementarity interactions between species is sharing resources via cross-feeding. This could positively influence growth, if for example some microorganisms are producing amino acids from gluconeogenic substrates while others produce them from glycolytic substrates[51]. In this example, microorganisms could obtain amino acids produced by one of their neighbors under high moisture, resulting in a more efficient (less expensive) community growth. Moreover, as CUE is a compilation of growth and respiration, CUE is only positively impacted by diversity if diversity influences growth more than respiration (Fig. 2 and Supplementary Fig. 10). Changes in abiotic factors could also alter the nature of species interactions by changing resource uptake rates[25] and/or requirements[52]. We observed a positive relationship between soil aggregation and growth, respiration and CUE within the microcosms incubated at low water content (Supplementary Fig. 11). This could indicate that under low moisture conditions, microbial community growth was more limited to the resources present at the aggregate level and therefore correlated to aggregate size. On the contrary, in moist soils no relationship was observed between soil aggregation and growth, respiration or CUE suggesting that in these soils microorganisms were not limited to the resources present at the aggregate level. Thus, low water content may have limited the extent of possible inter-species complementarity interactions which could explain the absence of positive relationship between diversity and CUE in

these soils. Alternatively, another possible mechanism is the additional costs due to desiccation stress[15], which could have impaired the positive relationship found in high moisture samples. Thus, the impact of microbial diversity on CUE is contingent upon abiotic conditions.

**Temperature and moisture effects on CUE.** Temperature[15,18] and water content[19,23] drive CUE in soil, and in accordance with previous studies[18], we measured lower CUE in microcosms incubated at higher temperatures ($t = 10.75$, df $= 172$, $P < 0.0001$). This decrease in CUE for communities incubated at higher temperature was associated with an increase in estimated rrN copy number (Supplementary Fig. 12), a high rrN copy number has been related to a lower growth efficiency[53], although this has not always been observed[22]. Moisture treatment showed no significant impact on either CUE ($t = -1.81$, df $= 161$, $P = 0.070$) or rrN (Supplementary Fig. 12). Given that microbial communities differed among the long-term temperature and moisture conditions (Table 1 and Supplementary Figs. 5–8) we simultaneously ran an additional incubation to evaluate the direct physiological response to short-term changes in temperature and moisture.

We measured CUE under all different abiotic combinations in a subset of microcosms (Fig. 3a). For this we selected all D0 microcosms grown under low water content. While long-term abiotic conditions are known to alter microbial community structure[26] the short-term shift in these conditions should induce physiological changes independent of community shifts[15]. We hypothesized CUE would decrease with increasing temperature[15,18]. However, the short-term increase of 10 °C in temperature significantly increased mass-specific respiration (Fig. 3b) and growth (Fig. 3c) to a similar degree (188% and 176%, respectively), and did not resulted in a significant CUE response (Fig. 3d) ($t = 0.36$, df $= 51$, *ns*). Garcia et al. showed that

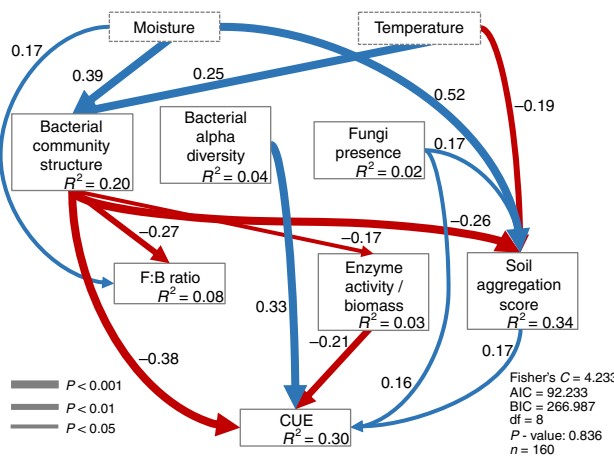

**Fig. 4 Structural equation model showing the relative influence of soil abiotic and biotic factors on CUE.** Significant paths are shown in blue if positive or in red if negative. Path width corresponds to degree of significance as shown in the lower left. The amount of variance explained by the model ($R^2$) is shown for each response variable, and measures of overall model fit are shown in the lower right. Bacterial community structure: axis 1 of NMDS; Bacterial alpha diversity: bacterial phylogenetic diversity index; Fungi presence: presence/absence of fungi; F:B ratio: 16S rRNA gene copy number $g^{-1}$ soil: ITS gene copy number $g^{-1}$ soil; Enzyme activity/Biomass: maximum activity recorded for Betaglucosidase/microbial biomass carbon. CUE: carbon use efficiency; Global goodness-of-fit: Fisher's C. Exact P values for each path coefficient are reported in Supplementary Table 1.

greater species richness is required to cope with warm temperatures to maintain growth[2]. Our D0 treatment was the most diverse, and these results suggest that its community was able to cope with the 10° increase in temperature without changing the respiration:growth relationship, and consequently without altering CUE.

Short-term changes in moisture had a stronger impact on CUE than temperature. An increase from 30 to 60% WHC elevated respiration and growth by 146% and 169%, respectively. The higher increase in growth compared to respiration after wetting the soil resulted in an 8% absolute increase in CUE (Fig. 3d). We hypothesize that higher growth was possibly due to higher nutrient availability when increasing the water content. Another possible explanation for moisture being a strong driver of CUE is the difference in water potential experienced by the microbial community at low compared to high water content[54] (Supplementary Fig. 13). A previous study showed that CUE decreased with drought duration[23], which could be associated with extra costs due to desiccation and the production of intracellular solutes or EPS, neither of which are captured by the DNA-based method of growth measurement used here.

**CUE as a function of interactions between biotic and abiotic drivers.** Model soil systems provide a unique platform for controlling specific biotic and abiotic components that play a major role governing soil processes, allowing the isolation of specific components from other confounding variables compared to natural soils. Thus, they can be used to increase understanding of major microbial ecology questions. We used structural equation modeling (SEM) to determine the degree to which the biotic components (fungal and bacterial diversity, community structure, fungal:bacterial ratio, potential extracellular enzyme activity and microbial-derived soil aggregation) mediate the influence of abiotic factors on CUE (Fig. 4 and Supplementary Figs. 14–16).

The model path structure was based on the supposition that abiotic drivers (water content and temperature) drive CUE directly, but also indirectly by impacting the biotic drivers of CUE (Supplementary Fig. 14). We used the SEM to test the following hypotheses: (1) distinct community structure will result in different community CUE; (2) bacterial diversity positively impacts CUE; (3) the extracellular enzymatic pool represents a cost to microbial growth efficiency and therefore has a negative effect on CUE; (4) the presence of fungi increases CUE, and (5) microbial-driven soil aggregation has a positive effect on CUE a proxy for substrate supply to cells. Overall, our model explained 30% of variance in CUE (Fig. 4).

Although temperature is commonly considered as a controlling variable for CUE[18], our structural equation model indicates that temperature and moisture influenced CUE only indirectly, and instead acted through the biotic components (Fig. 4). Bacterial-community structure and diversity were the strongest drivers of CUE. Bacterial diversity positively influenced CUE. However, we cannot make conclusions from the signal of the path coefficient between bacterial community structure and CUE because community structure is represented by the first axis of the non-metric multidimensional scaling (NMDS), which has an arbitrary direction. The other direct drivers of CUE were the presence of fungi, the extracellular enzymatic activity, and the soil aggregation. The potential extracellular enzymatic activity/ MBC was negatively related to CUE, supporting the idea that the enzyme poll represents a cost hindering growth efficiency as previously suggested[15,40,55]. We found fungal:bacterial ratio did not impact CUE in contrast to a previous study[41]. This difference might be due to the lower fungal:bacterial ratio in our artificial soil compared to natural soils[41] (Supplementary Fig. 17). Nonetheless, we observed a higher CUE in microcosms in which fungi were growing ("Fungi presence" component in the model). Accordingly, the $B_{only}$ treatment showed the lowest CUE values (Supplementary Fig. 9). However, fungal richness and community structure were not drivers of CUE (Supplementary Figs. 15 and 16), and we hypothesize that the 24 h of incubation for CUE measurements captured mainly bacterial growth as bacteria grow faster than fungi[56]. The positive effect of fungi presence on CUE indicates that some general fungal function is important for community growth efficiency (Fig. 4 and Supplementary Fig. 16). For instance, fungi could have provided sources of organic nitrogen to bacteria as evidenced by little to no N-acetylglucosaminidase (NAG) activity in $B_{only}$ microcosms (Supplementary Fig. 18). Thus, the impact of a microbial community on CUE can play out through a variety of mechanisms.

CUE is a composite variable of respiration and growth, which will depend on microorganisms physiology and environmental conditions. It is important to highlight that a substantial fraction of CUE variation remains unexplained in the model, meaning that other factors are important and were not captured here. Altogether these results highlight how changes in the abiotic environment (e.g., temperature and moisture) interact with community composition and diversity loss to impact community CUE.

## Conclusion

To face climate change we must understand how global environmental changes will impact soil C cycling. To our knowledge, this is the first study that actively manipulated microbial communities to explore how biotic and abiotic components interact to drive CUE in a soil system. Our results highlight that shifts in microbial communities can change CUE, and that the positive effect of microbial diversity on CUE is neutralized under dry conditions.

## Methods

**Model soil, inoculation, and incubation conditions.** We created a microbe and C-free soil to study biotic and abiotic drivers of CUE. The model soil consisted of 70% acid-washed sand, 20% muffled and acid-washed silt, and 10% calcium chloride-treated bentonite clay. After these fractions were combined, we determined the WHC of the soil. The model soil underwent three autoclave cycles with a minimum of 48 h intervals between each cycle to increase the chances of killing newly germinated spores. Each 20 g artificial soil microcosm was amended with 0.22 μM-filtered mixed deciduous leaf litter DOC (0.1 mg C g soil$^{-1}$), and 0.023 ml s g soil$^{-1}$ of a modified 2× VL55 base media with 5 μm-filtered yeast extract. We measured respiration in the fifteen days after addition of litter DOC and detected no respiration, which ensured that microcosms were sterile prior to inoculation.

Microbial communities were extracted from a temperate deciduous forest soil (Harvard Forest, Petersham, MA, USA, 42°30′00″N, 72°12′28″W) by shaking 2 g soil in 50 ml of a 5% sodium pyrophosphate solution. These microbial extracts were then manipulated with three different approaches prior to inoculation. The more concentrated extract, here named D0 inoculum, received in the form of liquid soil inoculum the equivalent of 0.004 g soil/g model soil. We then serially diluted the D0 inoculum up to 1000× to make the D1 treatment, and 100,000× to make the D2 treatment[37]. To exclude fungi and large bacterial cells, we took the D0 extract and filtered it through a 0.8 μm filter, generating the "bacteria only" treatment (B$_{only}$). To make the SF (spore-former enriched) treatment, we heated the soil to 120 °C for 1 h, and treated it with 1.5% phenol for 30 min. We also had non-inoculated microcosms which did not receive an inoculum (uninoculated controls; 20 samples). To reduce stochasticity during microbial communities extraction, all extractions were performed in duplicate and pooled before microcosms inoculation. Each replicate microcosm received soil from two pooled soil cores, with ten replicates per treatment. Microcosms were incubated at two different water content treatments (30 or 60% WHC) and two temperatures (15 or 25 °C) in a full factorial design (Fig. 1). Water potential measured by the HYPROP method[57] for 30 and 60% WHC is −418 and −31 kPa, respectively (Supplementary Fig. 13). Water content was adjusted to account for water evaporation during the weekly additions of substrate for the first 90 days of incubation, which consisted of 0.5 mg C g soil$^{-1}$ cellobiose and 0.05 mg N g soil$^{-1}$ ammonium nitrate solutions as sources of C and N, respectively. To allow for maximum utilization of nutrients, no substrate additions were done during the last 30 days of incubation. We also stirred the microcosms at the 91st day of incubation to ensure homogeneous substrate availability within the microcosms. Microbial community activity was monitored weekly by $CO_2$ flux measurements for the first month and then every 2 weeks thereafter. Because the SF treatment did not show respiration above abiotic controls until 6 weeks after inoculation, we let this treatment incubate for additional 6 weeks to account for equivalent time of microbial activity. However, the SF treatment at 15 °C and 30% WHC showed no measurable respiration by the end of the experiment and was therefore discarded from this study.

At the end of 120 days of incubation microcosms were harvested under sterile conditions. Each microcosm was sieved at 2 mm and allocated for different assays: 1.5 g for enzymatic assays, 1 g for gravimetric water content, 9 g for microbial biomass carbon (MBC) measurement, 2 g for aggregate formation, and 1.2 g for the $^{18}$O–H$_2$O-CUE assay. DNA was extracted from soils subject to the $^{18}$O–H$_2$O-CUE assay and used for the sequencing analysis and qPCR in addition to the $^{18}$O enrichment measurements. In addition to measuring the $^{18}$O–H$_2$O-CUE under the long-term incubation conditions, we performed a short-term incubation with a subset of the samples to evaluate how changes in the abiotic conditions (temperature and moisture) affect CUE independently of community shifts. To do this we selected the D0 diversity treatment at low moisture and both temperatures (15 and 25 °C) and applied all combinations of abiotic treatments.

**Microbial biomass carbon.** MBC was determined using the chloroform fumigation direct extraction method[58] with 0.05 M K$_2$SO$_4$. Briefly, three replicates with 1.5 g soil each received chloroform and K$_2$SO$_4$ buffer while other three replicates received only K$_2$SO$_4$ buffer. These samples were shaken for 30 min at 175 rpm, and left for 30 min at 4 °C to let soil particles settle. The supernatant was then filtered through an ashless Whatman 40 filter. Samples that received chloroform were subsequently bubbled for 20 min to volatilize any residual chloroform. Dissolved organic carbon (DOC) was measured colorimetrically, as DOC reduces Mn(III)-pyrophosphate it decreases the color of the solution in the presence of concentrated H$_2$SO$_4$[59] and MBC was calculated as DOC from soil extract receiving chloroform minus DOC from soil extract without chloroform. A concern with the direct method is that residual chloroform in the DOC extract could result in overestimating MBC. Here, we used the uninoculated controls (which should have zero MBC) to verify that DOC from soil extract that received chloroform minus DOC from soil extract without chloroform was zero. This confirms that no residual chloroform remained in the extract and MBC yields were not overestimated.

**Carbon use efficiency.** CUE was measured 48 h after microcosms harvest to allow soil to dry to add labeled or unlabeled water and measure CUE under the targeted moisture treatments. All samples had CUE measured under the long-term abiotic conditions for temperature and moisture to evaluate how distinct microbial communities influence CUE. In addition, to parse out the effect of different microbial communities and evaluate the impact of short-term changes in abiotic

conditions to CUE, we measured CUE from one diversity treatment under all possible abiotic combinations. For this we selected the D0 microcosms incubated at the lower moisture (30% WHC) and both temperatures (15 and 25 °C). Briefly, $^{18}$O–H$_2$O was added to 20% of the final water present to subsamples of each soil. Identical samples were prepared using $^{16}$O–H$_2$O as control for background heavy oxygen signature. All samples were then placed in sealed tubes for 24 h and the $CO_2$ produced during this time measured using an infrared gas analyzer (IRGA). The soil samples were stored at −80 °C until DNA extraction using the Qiagen Powersoil HTP kit. The resultant DNA was quantified using PicoGreen (Invitrogen), and its $^{18}$O enrichment was measured using TC/EA-IRMS (Delta V Advantage, Thermo Fisher, Germany) at the UC Davis Stable Isotope Facility. CUE was calculated as per Spohn et al.[39] but using a sample-specific conversion factor rather than the overall average due to large expected differences in MBC:DNA ratios across community types. Twenty two samples showed negative $^{18}$O-atom% excess resulting in negative growth values and were therefore excluded from the analysis.

**Quantitative real-time PCR (qPCR).** The abundance of total bacteria and total fungi was assessed by qPCR using 16S rRNA primers[60] and ITS primers[61], respectively. The abundance in each soil sample was based on increasing fluorescence intensity of the SYBR Green dye during amplification. An inhibition test performed by running serial dilutions of DNA extractions did not indicated inhibition of amplification prior to perform the qPCR. The qPCR assay was carried out in a 15 μl reaction volume containing 2 ng of DNA, 7.5 μl of SYBR Green PCR master mix (Qiiagen quantifast SYBR kit) and each primer at 1 μM. For each sample two independent qPCR assay were performed for each gene. The qPCR efficiencies for both genes ranged between 80 and 105%. qPCR values are reported as gene copy number g$^{-1}$ dry soil. These values were corrected to a genome counts basis using median values for ITS[62] copies and for bacterial[63] 16S ribosomal RNA operon copy number.

**Potential extracellular enzymatic activity.** Extracellular enzyme potentials were assayed for NAG and B-glucosidase (BG) as representative of N and C-cycling enzymes, respectively. Soil was kept at the long-term incubation temperature for 5 days following harvest before the enzyme assays were performed. BG activity was determined using (3000 μM) of 4-Methylumbelliferyl B-D-glucopyranoside and NAGase activity was assayed using (4000 μM) of 4-Methylumbelliferyl N-acetyl-B-D-glucosaminide, respectively. Incubation temperature was the sample-specific long-term temperature incubation. Plates were read immediately after substrate addition and after 2, 4, and 6 h with the excitation-emission wavelength pair of 350/450 nm on a Molecular Devices Spectramax M2 platereader. Potential activity was calculated as previously described[64]. Enzyme potential activities were normalized by MBC.

**Soil aggregation.** Aggregate distribution was determined with a modified water-stable protocol[65]. Briefly, for each sample duplicate 1 g air-dried soil samples were re-wet by capillarity to field capacity on a paper filter (11 cm diam.; Fisherbrand Paper Q5) in a 10 cm Petri dish. After 1 hr the aggregates were deposited on superposing sieves of 250, 106, and 53 μm (8 cm diam. each) and a 100 ml of DI water was used to flush the aggregates off the filter and through sieves. Aggregates remaining on each sieve were dried at 90 °C and then weighed. We determined the mean weight diameter (MWD) of aggregates[66]. The MWD is calculated as $\sum_{i=1}^{n} \overline{X}_i W_i$ where $\overline{X}$ is the average sieve size for each fraction and $W$ is the weight recovered in that fraction[66]. To account for microbial-derived aggregate formation, MWD of non-inoculated control microcosms for each specific abiotic condition were subtracted from samples MWD; the resulting value is herein named "Aggregation Score."

**Total C and N.** The soil used for measuring gravimetric water content was dried under 65 °C to constant weight, ground and analyzed for total C and N using elemental analysis (Leco Elemental analyzer).

**Bacterial and fungal diversity.** An aliquot of the same DNA extracted for CUE estimates was used to perform 16S rRNA gene and ITS region tagged amplicon sequencing using Ilumina MiSeq platform using protocols established by the Earth Microbiome Project[67] at the Argonne National Laboratory (Supplementary methods 1). Raw sequences from amplicon sequencing were quality filtered, merged and clustered to generate OTU's at 99 and 97% sequence similarity for bacteria and fungi respectively using QIIME2 pipeline[68]. Diversity matrices were calculated on 30,000 reads for bacteria and 1000 reads for fungi, respectively using the vegan[69] R library. Greengenes (version 13.8)[70] and UNITE (version 01.12.2017)[71] were used for the taxonomy assignment for bacteria and fungi, respectively.

**Statistical analysis.** Statistical analysis were performed in R statistical software (version 3.5.2), using the agricolae[72] and vegan[69] libraries. Normality of each variable was tested and log transformed if needed. Outliers were detected by verifying if an observation was outside the 1.5 × inter quartile range for the first and third quartiles. CUE has a biological maximum estimated about 80%[15]. If diversity relates to CUE, we were interested to evaluate if it saturates within the diversity

level observed in this study. The relationship between diversity and CUE was evaluated first by comparing the log and saturating exponential curves, and then upon seen these underpredicted CUE at high diversity we performed a break point analysis. The break point analysis was performed using a piecewise regression approach with the segmented package[73]. Significant differences between treatments were determined by analysis of variance and post hoc Tukey test. The impact of short-term changes in abiotic conditions on CUE, growth and respiration was evaluated with linear mixed effects model with microcosm as the random effect using the nlme package[74], and ANOVA type III to correct for unbalanced design because we lost six samples ($n = 72$ instead of 80).

NMDS of the UNIFRAC distance matrices (weighted) were used to describe bacterial community structure. Fungal NMDS used hellinger distance. We tested the effect of treatments (inoculation treatment and abiotic conditions) with Adonis and PERMANOVA in the vegan package[69].

We used SEM to test direct and indirect effects of abiotic and biotic parameters on CUE. The hypothesized path structure was based on the proposition that abiotic drivers (water content and temperature) drive CUE directly, but also indirectly by impacting the biotic drivers (Supplementary Fig. 14). Specifically, our hypothesis were: (1) microbial alpha diversity and community structure were driven by abiotic factors; (2) microbial alpha diversity and community structure are driving the extracellular enzymatic activity, the fungal:bacterial ratio and aggregation score; (3) fungal:bacterial ratio drives the aggregation score and (4) moisture, temperature, microbial alpha diversity and community structure, fungal:bacterial ratio, extracellular enzyme activity, and aggregation score drives CUE. Because we observed a nonlinear relationship between bacterial diversity and CUE, we log transformed the bacterial data for the SEM. A partial bivariate correlation was identified between bacterial diversity and bacterial community structure and added to the model as it significantly improved model fit ($P < 0.05$). The SEM model path fit was performed using the piecewiseSEM package[75], which allows for distinct relationships (e.g., linear, binomial, etc.) between the variables within the model[76]. We kept the model that explained the most variation in CUE, and had a nonsignificant Chi-squared test ($P > 0.05$), low Akaike Information Criterion (AIC) and high Comparative Fit Index (CFI > 0.9). Figures were made using the ggplot2 package[77].

**Data reproducibility**. Model soil systems provide a platform for controlling specific biotic and abiotic components that play a major role governing soil processes, allowing the isolation of specific components from other confounding variables compared to natural soils. Here, we used a model soil to evaluate how microbial diversity, community structure, moisture, and temperature drive CUE. Although this experiment was performed once we used a relatively high number of biological and technical replicates for the different assays to increase reproducibility. To reduce stochasticity during soil microbial communities extraction, all extractions were performed in duplicate and pooled before microcosms inoculation. Moreover we performed technical duplicates for DNA extraction. Thus, every microcosms had 2 subsamples receiving $^{18}O–H_2O$ and 2 subsamples receiving $^{16}O–H_2O$, adding up to 800 DNA extractions. DNA extractions were quantified and subsamples pooled prior to sending the samples to the Stable Isotope Facility. The MBC assay was performed with three technical replicates; the aggregate formation assay was performed with two technical replicates; qPCR assays were done in duplicate and with two independent assays performed for each gene for each sample and extracellular enzyme activities had seven replicates for each enzyme type and concentration.

**Reporting summary**. Further information on research design is available in the Nature Research Reporting Summary linked to this article.

## Data availability
The data supporting the findings presented here are available from the corresponding authors on request and from the https://osf.io/qmf8z/Open Science Framework Repository. The sequencing data are available in the NCBI repository with the identifiers https://www.ncbi.nlm.nih.gov/bioproject/PRJNA556439PRJNA556439 and https://www.ncbi.nlm.nih.gov/bioproject/PRJNA556522PRJNA556522 for bacteria and fungi, respectively.

## Code availability
The R code supporting the findings presented here is available from the corresponding authors on request and from the Open Science Framework Repository (https://osf.io/qmf8z/).

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

## Acknowledgements

Funding for this project was provided by the Department of Energy grant DE-SC0016590 to K.M.D. and S.D.F., and an American Association of University Women Dissertation fellowship to G.P. We would also like to thank Stuart Grandy and Kevin Geyer for the fruitful discussions and Mary Waters, Courtney Bly and Ana Horta for their help with samples processing.

## Author contributions

L.A.D.H., G.P., and K.M.D. designed the experiment, L.A.D.H conducted the experiment, sample processing, data analysis, and wrote the paper. All authors, G.P., X.J.A.L., S.D.F., J.M.M., and K.M.D. discussed the results, and contributed to write the paper.

## Competing interests

The authors declare no competing interests.
