## [Peer Review File · Nature Communications]

Reviewers' Comments:

Reviewer #1:

Remarks to the Author:

Domeignoz-Horta et al. present a work on the effects of biotic- and abiotic factors on carbon use efficiency (CUE). They generate a variety of novel soil microbial communities with a model soil and compare microbial CUE under different conditions of warming and drought. They found abiotic factors indirectly influence CUE through changes in microbial community diversity and composition, in particularly for soil bacterial community structure. Their results also indicate diversity-CUE relationships are context dependency, which can be modulated by soil moisture. As to my knowledge, this is the first empirical work using lab incubation to explore diversity-CUE relationships by considering several scenarios of climate change. This work is very interesting and would have great impact on the field of soil microbial ecology and climate change. However, there are some issues that need to address before publication.

Major issues:

1. Statistical analyses:

- (1) potential outliers of bacterial phylogenetic diversity under 30% WHC (values > 9; Figure 2) could influence of the bivariate association between bacterial phylogenetic diversity and CUE. It's unclear whether the outliers have been considered in the processes of data analyses.
- (2) non-linear relationships are shown in Figure 2. Again, it's unclear how the non-linear relationships are addresses in the structural equation model.
- (3) hypotheses presented in Figure 4 are too general. For example, what are the expected effects of community structure on CUE? The expected effects also apply for the other paths and need to justify but are not explicitly addressed in this version of manuscript. The processes for the model selection are also not clear. I cannot tell why the direct paths from moisture and temperature to CUE are not included.

2. Result interpretation

- (1) Lines 200-201: This is a good point that nutrient availability and water content jointly influence microbial growth. But it seems like an uncompleted point. Why should I expected nutrient availability should contribute to microbial growth regarding to the incubation soil system? Please provide more information to explain this point.
- (2) Lines 221-223: Given enzyme activity measured in this study is potential activity (a measure is usually confounding with enzyme activity and enzyme production, the enzyme assays used cannot tease one from another), I don't think enzyme production is the most likely reason to explain the negative effect of enzyme activity on CUE because this study does not exactly measure enzyme production.
- (3) Line 206: In this section, I am looking for the explanation for the negative effect of bacterial community structure on CUE, but I could not find it. This concern is consistent with the issue with the hypotheses of the SEM. I suggest the authors carefully present their hypotheses and provide reasonable explanation on the corresponding findings if possible.

3. Implications:

Lines 1-2: Microbial diversity driving CUE is based on the results of lab incubation. I am wondering how general I can expect for this work what are the implications in real world. These issues might be worth to discuss in the main text of the manuscript. In addition, microbial diversity is created by different experimental manipulations (D0, D1, D2, Bonly, SF). CUE could be also influenced by the experimental manipulation. That is, there is one alternative that has not consider in this manuscript, i.e., microbial diversity could co-vary with CUE, rather than microbial diversity drives CUE. Finally, the experimental design could also allow to quantify the unique contributions of those abiotic- and biotic factors on CUE, as well as the fungal and bacterial contributions on CUE, it might be good to explicitly present these potential mechanisms.

Minor issues:

Lines 18-20: This result need to be re-considered after excluding the outliers in low moisture.

Line 23: Why would I expect climate change would have negative effect on the provision of ecosystem functions? For instance, climate warming might increase forage production.

Line 39: It might be good to justify why should I care about temperature and water availability given there are several abiotic factors could influence CUE simultaneously. Is that because the two factors can be easily controlled in the lab? The rationale is unclear here.

Line 82: I know there are 200 samples in total considered. But it's hard to tell how many samples they used for the specific analyses. For example, Line 284, the SF treatment at 15 oC and 30% WHC is excluded. Will this influence the results or interpretation? Line 318, how many negative samples are excluded? It would be good if the authors provide the exact sample size for their statistical analyses.

Lines 105-106: I don't understand the median fungal richness was zero. That means negative fungal richness is also observed, right?

Lines 114-115: this sentence does not seem like a good fit to the results section. I am wondering how I can understand the same density reached.

Lines 125-127: What does this mean the level of diversity? It might does not make sense to compare this study with the other studies.

Lines 150-152: Is this a common level of community diversity (<10 species) in natural ecosystems? Sounds like the review is based on biodiversity experiment.

Line 258: typo in base media? with.

Line 268: 120oC?

Lines 273-274: I still do not follow the rationale of the experimental design. Why 30 or 60% WHC and 25 or 25oC? What do the experimental settings mean in real world?

Line 280: added substrate no substrate?

Line 299-301: I don't understand this sentence.

Line 313 and 326: Qiagen?

Line 332: I don't understand why NAG and BG are related to P cycling.

Line 357: Not clear. What do you mean same DNA extracted for CUE estimates?

Line 387: Comparative Fit Index? Are you sure you can obtain this index from piecewiseSEM?

Line 466: Author names are missing.

Line 478 and 504: Journal names are missing. Please check though the references.

Figure 1: delete one between.

Figure 2: What do (a), (b) and (c) really represent in this figure?

Figure 3: grown at grown?

I hope my comments and suggestions would be helpful for the revision of the manuscript.

Xin Jing
2020.01.14

Reviewer #2:

Remarks to the Author:

The manuscript by Domeignoz-Horta and co-workers investigates how microbial community metabolism (as measured by C-use efficiency (CUE) = ratio of growth over substrate consumption) changes in response to different abiotic conditions (temperature, soil moisture) and microbial community diversity or composition. This work is timely, as most other papers on this topic focus on abiotic conditions as drivers of microbial metabolism, and less on microbial community characteristics. I find the experimental design adequate, but the description of the approach is not always clear. I also have several relatively minor comments.

- Methods: I would clarify in the Methods why a short-term experiment is conducted – it now reads as if the short-term experiment is just in addition to the long-term one. The motivation is there (around L307), but somehow hard to follow. Moreover, it is not clear why the dry D0 treatment

was used for the short-term incubations. Also, see comments below regarding some regression analysis – data are clustered and a regression model seems to show strong bias at high diversity levels.

- Interpretation of aggregate-CUE relations: aggregation is positively correlated to CUE in the dry treatment, but the explanation provided in L167 that growth was limited by resources in aggregates is not convincing. Were growth and respiration rates correlated to aggregation? Was the C or nutrient content of aggregates measured to support this explanation?

- Figure 1 shows the soil samples with apparent microbial growth – this makes me wonder if the biomass and C fluxes measured in the long-term incubations are comparable to those that would be measured in natural soils. I realize this is a study based on idealized conditions, but it is relevant to discuss similarities and differences compared to the real world.

Other comments

L85: at this point in the manuscript, it is not clear how aggregation is connected to substrate supply –some explanations are needed

L102: in the Methods (P13, L265), dilution is reported to be of four (not five) orders of magnitude

L138: does “scaled” mean “increased”?

L153: perhaps “is consistent with” instead of “corroborates to”?

L157: this hypothesis is a bit far fetched and not testable here – especially the link to CUE – because diversity is manipulated but not complementarity. The hypothesis is plausible, but some more explanations of why CUE should increase when species are complementary are needed

L179: typo “effect on CUE”

L181: I do not see the connection between these sentences. The premise is that in the long-term it is difficult to separate abiotic vs. biotic drivers on CUE. As explained elsewhere in the manuscript, long-term and short-term experiments tell different parts of the story, but a short-term experiment won't help disentangling the long-term abiotic vs. biotic effects on CUE

L187: why selecting the dry D0 treatment?

L247: or the other way around? From previous statements and structural equation modelling, it seems that abiotic factors are acting indirectly on microbial metabolism, the effects being mediated by microbial community characteristics – so it is the abiotic effects that are masked by the biotic ones

L248: this work shows that microbial diversity matters, but why “more than previously believed”? this seems a bit of an overstatement...

L264: shouldn't the inoculated amount be expressed as a volume of inoculum solution per gram of model soil? Or was 'real' soil mixed with the model soil? Please clarify

L275: higher WHC should correspond to higher (less negative) water potential

L305: I would add a short sentence to explain that in addition to the long-term incubations, you also want to perform short-term incubations to test abiotic effects on rates independent of community changes. It is stated later, but I found this transition not very clear

L337: typo “establish”

L334: typo “capillarity”

L348: the sentence starting here is convoluted and hard to follow

L466: reference missing author list

Figure 1: please check dilution value – 5 or 4 orders of magnitude?

Figure 2 (also some figures in SI): data points are clustered at low diversity levels, and variance does not seem homogeneous – how were these inhomogeneities in the data treated when conducting the regression analysis? Also, the CUE function in the bottom right panel seems to completely miss the high CUE values at high diversity, suggesting that the model chosen for the fitting is not appropriate

Figure 3: I would suggest highlighting the bars in the box plots that correspond to unchanged abiotic conditions in the long- and short-term incubations. These are reference points for the other bars

We are grateful to the reviewers for their careful reading and comments on our article. Their comments helped us to improve the article. Please find below our answer to every comment. As requested, we highlighted in the article (in orange) the changes that were made in the text. When appropriate, we refer to the line numbers in the newly revised version of the manuscript.

Reviewers' comments:

Reviewer #1 (Remarks to the Author):

Domeignoz-Horta et al. present a work on the effects of biotic- and abiotic factors on carbon use efficiency (CUE). They generate a variety of novel soil microbial communities with a model soil and compare microbial CUE under different conditions of warming and drought. They found abiotic factors indirectly influence CUE through changes in microbial community diversity and composition, in particularly for soil bacterial community structure. Their results also indicate diversity-CUE relationships are context dependency, which can be modulated by soil moisture. As to my knowledge, this is the first empirical work using lab incubation to explore diversity-CUE relationships by considering several scenarios of climate change. This work is very interesting and would have great impact on the field of soil microbial ecology and climate change. However, there are some issues that need to address before publication.

Major issues:

1. Statistical analyses:

(1) potential outliers of bacterial phylogenetic diversity under 30% WHC (values > 9; Figure 2) could influence of the bivariate association between bacterial phylogenetic diversity and CUE. It's unclear whether the outliers have been considered in the processes of data analyses.

We share the same concern with the reviewer concerning the outliers. To verify if the relationships in Figure 2 holds, we used two approaches: 1) We deleted the outliers and verified that the relationship still holds; 2) to verify that the relationship is not driven by only a few points, we deleted a third of the data randomly and ran the analysis again, confirming that we still obtained a significant relationship between diversity-CUE in wet soils and no relationship in dry soils. Thanks for raising this issue. We have added a line to describe how we treated outliers in the methods section (line 403-404).

(2) non-linear relationships are shown in Figure 2. Again, it's unclear how the non-linear relationships are addresses in the structural equation model.

We added an explanation on how we treated the non-linear relationship between diversity and CUE in the structural equation model (SEM) (lines 426-429). Because we observed a non-linear relationship between diversity and CUE, we log transformed the values for the bacterial phylogenetic diversity index before input into the SEM.

There are different methods to perform SEM. The method we used allows us to estimate different relationships between the different variables in the model. The first method used is lavaan which estimates the relationships among the variables based on attempts to reproduce a single variance-covariance matrix. This method is described as "global estimation" because the variance-covariance matrix captures relationships among all variables in the model at once. Because this method does not allow variables to have different relationships within the model, we used a different approach called piecewise SEM (Shipley et al., 2000). The piecewiseSEM method estimates separately the relationship between the variables ("one by one or piece by piece") and strings together their inferences after. This method allows to recognize distinct

relationship between the individual variables. This means that while some variables might have a linear relationship between them others can have non-linear, Poisson-distributed, etc. The piecewiseSEM approach is for us a more appropriate approach to be used in our study.

Reference: Shipley, Bill. "A new inferential test for path models based on directed acyclic graphs." *Structural Equation Modeling* 7.2 (2000): 206-218.

(3) hypotheses presented in Figure 4 are too general. For example, what are the expected effects of community structure on CUE? The expected effects also apply for the other paths and need to justify but are not explicitly addressed in this version of manuscript. The processes for the model selection are also not clear. I cannot tell why the direct paths from moisture and temperature to CUE are not included.

Thanks for noticing this caveat on our explanation. We extended the explanations in the results & discussion session (line 236-240 and the methods session (419-421). We showed in our previous study (Pold et al., 2020) that individual taxa have distinct CUE. Here, we are hypothesizing that community composition matters for CUE. Thus, if the microbial community shifts, it should drive changes in CUE and this is what our SEM shows. We did not make a more explicit hypothesis regarding the positive or negative effect of community structure on CUE because we used the axis 1 of the NMDS as a proxy for community structure and the direction of the NMDS axis is arbitrary. Therefore, we are not able to draw further hypotheses or conclusions from the signal of the path coefficient going from community structure to CUE. Admittedly, we can only affirm that community structure drives CUE.

The paths for moisture and temperature to CUE were added in the hypothesized model (Supplementary Fig. 14), but those paths did not significantly drive CUE. Our model shows that the abiotic factors impacted the communities and it was the communities that directly affected CUE. We added more information to the SEM description in the methods section (line 418-426).

Reference:

Pold G, Domeignoz-Horta LA, Morrison EW, Frey SD, Sistla SA, DeAngelis KM. 2020. Carbon use efficiency and its temperature sensitivity covary in soil bacteria. *mBio* 11:e02293-19. <https://doi.org/10.1128/mBio.02293-19>.

2. Result interpretation

(1) Lines 200-201: This is a good point that nutrient availability and water content jointly influence microbial growth. But it seems like an uncompleted point. Why should I expected nutrient availability should contribute to microbial growth regarding to the incubation soil system? Please provide more information to explain this point.

We extended the explanation regarding this hypothesis (lines 173-188). We observed a positive relationship between aggregate size and growth, respiration and CUE in dry soils but not wet soils (Supplementary Fig. S11). This suggests that substrate availability was limited in drier soils which could indicate limited physical connectivity between soil pores. In wetter soils when more water was added to soil it allowed connectivity between soil aggregates and the relationship between aggregate size and growth is not present anymore.

(2) Lines 221-223: Given enzyme activity measured in this study is potential activity (a measure is usually confounding with enzyme activity and enzyme production, the enzyme assays used cannot tease one from another), I don't think enzyme production is the most likely reason to

explain the negative effect of enzyme activity on CUE because this study does not exactly measure enzyme production.

This is a good point highlighted by the reviewer. We agree that we are measuring the potential activity of the enzyme pool and not enzyme production. We have changed our language to potential enzyme activity in the text. For the SEM we are using the enzyme pool normalized to microbial biomass carbon, to capture the size of the enzyme pool in relation to the community size, which is an attempt to capture the energy a community has put into the enzymatic pool (Malik et al., 2019). We understand there is no perfect answer to this issue, so we are using what we considered to be the more meaningful estimates in an ecological and biological context.

References:

Malik, A. A., Puissant, J., Goodall, T., Allison, S. D. & Griffiths, R. I. Soil microbial communities with greater investment in resource acquisition have lower growth yield. *Soil Biology and Biochemistry* 132, 36–39 (2019)

(3) Line 206: In this section, I am looking for the explanation for the negative effect of bacterial community structure on CUE, but I could not find it. This concern is consistent with the issue with the hypotheses of the SEM. I suggest the authors carefully present their hypotheses and provide reasonable explanation on the corresponding findings if possible.

The hypotheses tested with the SEM have been incorporated in the article (lines 236-240). We previously showed that distinct bacterial isolates have intrinsically different CUE (Pold et al., 2020). Thus, our hypothesis regarding the community structure-CUE relationship is that a change in community structure should drive a change in CUE. To include the community structure in the SEM, we used the first axis of the NMDS. The axis values from an ordination are arbitrary (specific position of a sample compared to all other samples). Therefore, we are not able to draw further conclusions from the signal of the path coefficient going from community structure to CUE. We can only conclude that a change in community structure influences CUE. We have added the explanation in lines (245-249).

3. Implications:

Lines 1-2: Microbial diversity driving CUE is based on the results of lab incubation. I am wondering how general I can expect for this work what are the implications in real world. These issues might be worth to discuss in the main text of the manuscript. In addition, microbial diversity is created by different experimental manipulations (D0, D1, D2, Bonly, SF). CUE could be also influenced by the experimental manipulation. That is, there is one alternative that has not consider in this manuscript, i.e., microbial diversity could co-vary with CUE, rather than microbial diversity drives CUE. Finally, the experimental design could also allow to quantify the unique contributions of those abiotic- and biotic factors on CUE, as well as the fungal and bacterial contributions on CUE, it might be good to explicitly present these potential mechanisms.

This is an important comment, and we thank the reviewer for giving us the opportunity to address it. In light of this comment, we have edited the manuscript to make clear how our model system applies to the real world of soil microbial communities (lines 122-126 and 151-166).

This work used a model soil system, which is a simplified system compared to natural soils. However, compared to other studies that are manipulating diversity to study the relationship between diversity and ecosystem function, we obtained a relative high number of

species in our model soil (ranging from 10-100). For example, a review by Nielsen et al. (2011) showed that there was a positive relationship between diversity and C cycling in low diversity experiments (with less than 10 species) while studies with more species recorded less frequently a positive relationship between diversity and C cycling. A second important point is that we manipulated communities in a soil-like matrix, and we are not using a liquid culture medium to evaluate the diversity-ecosystem function relationship. A number of studies used liquid culture medium (Bell et al., 2005, Garcia et al., 2018, Nielsen et al., 2011) to infer how diversity of soil communities impact function. By using a soil-like matrix we think we are being more realistic of the processes occurring in soil and due to that, we were able to capture the interaction between abiotic and biotic factors that would not be evaluated when using a liquid cultivation method.

Regarding the possibility that diversity could co-vary with CUE, in a previous study, we showed that CUE co-varies among bacterial taxa (Pold et al., 2020). In that study we measured CUE of 26 bacterial isolates from seven families, grown on one of four different substrates. Because CUE is the compilation of respiration and growth, we interpret CUE as a microbial component that helps understand microbial physiology. In this present study, we manipulated microbial diversity and not CUE. We think that it is more meaningful to evaluate how microorganisms drive CUE at not vice-versa.

The question regarding the contribution of different factors to CUE: We built the structural equation model to parse out the drivers of CUE. And to evaluate the impact of short-term changes in abiotic conditions, we incubated a subset of soils from the long-term experiment under all different abiotic conditions (Fig. 3). This allowed us to parse out the influence of temperature and moisture for respiration, growth and CUE.

References:

- Bell, et al., 2005. The contribution of species richness and composition to bacterial services. *Nature* 436, 1157–60.
- Garcia, F. C., Bestion, E., Warfield, R. & Yvon-Durocher, G. Changes in temperature alter the relationship between biodiversity and ecosystem functioning. *Proceedings of the National Academy of Sciences* 115, 10989–10994 (2018)
- Pold G, Domeignoz-Horta LA, Morrison EW, Frey SD, Sistla SA, DeAngelis KM. 2020. Carbon use efficiency and its temperature sensitivity covary in soil bacteria. *mBio* 11:e02293-19. <https://doi.org/10.1128/mBio.02293-19>.
- Nielsen et al., 2011. Soil biodiversity and carbon cycling: a review and synthesis of studies examining diversity–function relationships. *European Journal of Soil Science*, 62, 105–116 <https://doi.org/10.1111/j.1365-2389.2010.01314.x>

Minor issues:

Lines 18-20: This result need to be re-considered after excluding the outliers in low moisture.

This has been verified and this conclusion remains valid (lines 17-19).

Line 23: Why would I expect climate change would have negative effect on the provision of ecosystem functions? For instance, climate warming might increase forage production.

This is an interesting aspect addressed by the reviewer. While some regions in the world can observe a negative impact on ecosystem function, other regions might observe a positive effect

in specific functions like biomass production in boreal regions. However, the IPCC 2014 and most scientific reports (reviewed by Cavalcanti et al., 2019) suggest more negative influences. Therefore, we chose to keep the text as it is in line 23.

Line 39: It might be good to justify why should I care about temperature and water availability given there are several abiotic factors could influence CUE simultaneously. Is that because the two factors can be easily controlled in the lab? The rationale is unclear here.

Thank you, we modified the text according to the reviewer suggestion. We decided to study these factors because global change is causing changes in temperature and precipitation (which will impact water availability in soils) and these are also variables that are easily controlled in soils. Modified text in line 39-40.

Line 82: I know there are 200 samples in total considered. But it's hard to tell how many samples they used for the specific analyses. For example, Line 284, the SF treatment at 15 °C and 30% WHC is excluded. Will this influence the results or interpretation? Line 318, how many negative samples are excluded? It would be good if the authors provide the exact sample size for their statistical analyses.

Thank you, we added the specific number of samples we used in each statistical analysis (in each figure legend). We deleted the spore forming (SF) treatment at 15°C and 30% WHC because this treatment showed no measurable respiration. This does not influence the results or interpretation because we treated our dataset as a continuous variable to evaluate if diversity impacts CUE (Fig. 2) and the same applies to the structural equation model. We also added the numbers of samples deleted in line 356-357 because we could not measure growth (18 samples).

Lines 105-106: I don't understand the median fungal richness was zero. That means negative fungal richness is also observed, right?

The objective of this treatment was to exclude fungi and have predominantly bacteria. We modified the sentence to improve clarity (lines 103-105). This means that in samples for the “Bonly” treatment (“bacteria predominantly”), most samples showed zero fungal OTUs because there were no fungi growing in this treatment for most of the replicates.

Lines 114-115: this sentence does not seem like a good fit to the results section. I am wondering how I can understand the same density reached.

We agree with the reviewer and we deleted this sentence. Thank you. Regarding reaching the density: because all microcosms received the same amount of substrate, even if you start with a different number of cells per microcosm in the beginning, they should reach the same cell density at the end of the incubation because the resource availability was the same. This has been previously showed by Philippot et al. (2013) using the same diversity removal approach that we used here.

Reference:

Philippot, L. et al. Loss in microbial diversity affects nitrogen cycling in soil. ISME J, 1609–1619 (2013).

Lines 125-127: What does this mean the level of diversity? It might does not make sense to compare this study with the other studies.

We agree with the reviewer comment discussing what are the implications of this work for real soils. These lines are an attempt to put in perspective our work in relation to other research that has evaluated the relationship between diversity and ecosystem function. We agree that our model soils are a simplified system to study natural soils, so we think it is important to remember that natural soils are more diverse than this model soil. However our system has a higher number of species compared to other studies that have tried to evaluate the same questions and we are using a soil-like matrix. We have revised the text to improve clarity (lines 122-126)

Lines 150-152: Is this a common level of community diversity (<10 species) in natural ecosystems? Sounds like the review is based on biodiversity experiment.

We agree with the reviewer that such low diversity is not representative of natural ecosystems. This is a challenge for empirical experiments which aim to manipulate diversity and keep other variables constant. They will always represent a simplification of reality. Nielsen et al. (2011) suggests that a positive relationship between diversity and C-cycling functions is mostly observed in low diversity treatments, in the range of tens of species. In our study our lowest diversity treatment has more than 10 species and we still observed a positive effect of diversity on C-cycle functions. This could be due to the fact that we used a soil-like matrix which adds complexity in the system requiring a higher number of organisms to maintain the studied C-functions. We have modified these lines to improve clarity (lines 151-166).

Line 258: typo in base media? with.

It was been corrected. Thank you.

Line 268: 120oC?

It has been corrected. Thank you.

Lines 273-274: I still do not follow the rationale of the experimental design. Why 30 or 60% WHC and 25 or 25oC? What do the experimental settings mean in real world?

First, we should clarify that 60% and 30% Water Holding Capacity (WHC) is equivalent to about 20% and 10% soil moisture in these soils. We used 60% WHC because this is considered to be a condition optimum for microbial growth. We chose the other condition (30%) because we wanted to simulate drought, and in a previous experiment, we didn't observe growth at 20 and 10% WHC. Regarding temperature, we chose 15°C and 25°C as temperatures that are within the range often recorded during the growing season in temperate climates, but also temperatures of tropical soils. We limited ourselves to two water availability and temperature treatments because when using a full factorial design with five different diversity manipulation treatments, this equals 200 microcosms (plus 20 abiotic control microcosms). From a practical point of view, we could not manage more microcosms in a single experiment.

Line 280: added substrate no substrate?

We modified the sentence to improve clarity (lines 308-311). Here what we mean is: substrate was added during the three months and on the final month, no substrate was added to the microcosms to account for maximum utilization of previously added substrate.

Line 299-301: I don't understand this sentence.

We modified the description to improve clarity (lines 335-339). The most commonly used protocol to measure microbial biomass carbon (MBC) chloroform is used to fumigate the soil (Vance et al., 1987). In a preliminary experiment, we validated an alternative method (Setia et al., 2012) where chloroform is directly added to the soil to lyse the cells. A concern with the direct method is that residual chloroform in the DOC extract could result in overestimating MBC. Here we used the uninoculated controls (which should have zero MBC) to verify that DOC from soil extract that received chloroform minus DOC from soil extract without chloroform was zero. This confirms that no residual chloroform remained in the extract and MBC yields were not overestimated.

References:

Vance, E. D., Brookes, P. C. & Jenkinson, D. S. An extraction method for measuring soil microbial biomass C. *Soil Biol. Biochem.* 19, 703–707 (1987).

Setia, R., Verma, S. L. & Marschner, P. Measuring microbial biomass carbon by direct extraction – comparison with chloroform fumigation-extraction. *European Journal of Soil Biology* 53, 103 – 106 (2012).

Line 313 and 326: Qiagen?

Thank you, this has been corrected (line 354).

Line 332: I don't understand why NAG and BG are related to P cycling.

Sorry, it was a typo. We excluded the reference to P cycling.

Line 357: Not clear. What do you mean same DNA extracted for CUE estimates?

We have modified the sentence to make it clearer (lines 320-326). The method used here to measure CUE requires DNA extraction (Spohn et al., 2016). Thus, we used a fraction of this same DNA extractant used for the ¹⁸O-H₂O-CUE for sequencing the bacterial and fungal communities growing in this artificial soil.

Reference:

Spohn, M. et al. Soil microbial carbon use efficiency and biomass turnover in a long-term fertilization experiment in a temperate grassland. *Soil Biology and Biochemistry* 97, 168–175 (2016).

Line 387: Comparative Fit Index? Are you sure you can obtain this index from piecewiseSEM?

The Comparative Fit Index (CFI) considers the deviation from a “null” model. In a null model, all the covariances are set to zero. Comparative fit index (CFI) is one of the fit statistics that piecewiseSEM offers (Hu and Bentler, 1999).

Reference:

Hu, Litze; Bentler, Peter M. Cutoff criteria for fit indexes in covariance structure analysis: Conventional criteria versus new alternatives. *Structural Equation Modeling: A Multidisciplinary Journal*. 6: 1–55 (1999).

Line 466: Author names are missing.

Thank you, we corrected the problems with the reference.

Line 478 and 504: Journal names are missing. Please check though the references.

Thank you, we corrected the problems with the reference.

Figure 1: delete one between.

Thank you for identifying this typo in the legend.

Figure 2: What do (a), (b) and (c) really represent in this figure?

In Figure 2, a, b and c represent the panels corresponding to growth, respiration and CUE, respectively. We corrected the legend.

Figure 3: grown at grown?

Thank you for identifying this typo!

I hope my comments and suggestions would be helpful for the revision of the manuscript.

Xin Jing
2020.01.14

Reviewer #2 (Remarks to the Author):

The manuscript by Domeignoz-Horta and co-workers investigates how microbial community metabolism (as measured by C-use efficiency (CUE) = ratio of growth over substrate consumption) changes in response to different abiotic conditions (temperature, soil moisture) and microbial community diversity or composition. This work is timely, as most other papers on this topic focus on abiotic conditions as drivers of microbial metabolism, and less on microbial community characteristics. I find the experimental design adequate, but the description of the approach is not always clear. I also have several relatively minor comments.

- Methods: I would clarify in the Methods why a short-term experiment is conducted – it now reads as if the short-term experiment is just in addition to the long-term one. The motivation is

there (around L307), but somehow hard to follow. Moreover, it is not clear why the dry D0 treatment was used for the short-term incubations. Also, see comments below regarding some regression analysis – data are clustered and a regression model seems to show strong bias at high diversity levels.

Thank you for pointing this out. We modified the description in the methods to make this clearer (lines 343-347). Regarding why we chose the dry DO treatment, we selected it somewhat randomly. Logistically, we could manage only one treatment due to the high number of samples needed. Taking one diversity treatment at two temperatures (20 samples) and applying all combinations of abiotic treatment resulted in 80 samples. We could not measure growth in eight samples, this is why we have 72 samples as indicated on the figure legend and in the methods session (line 413). Please see below our answer regarding the statistical analyses.

- Interpretation of aggregate-CUE relations: aggregation is positively correlated to CUE in the dry treatment, but the explanation provided in L167 that growth was limited by resources in aggregates is not convincing. Were growth and respiration rates correlated to aggregation? Was the C or nutrient content of aggregates measured to support this explanation?

We expanded our explanation on our hypothesis that soil aggregation in this model soil is a proxy for substrate supply (lines 182-190). We also added to Supplementary Fig 11 the growth and respiration relationships with soil aggregation. Growth and respiration are also positively correlated to the aggregation score under dry conditions. This suggests that in dry soils there was substrate limitation due to lower diffusion rates that was not present in moist soils. We did not measure the C content of aggregates. We aim to do this in a future experiment.

- Figure 1 shows the soil samples with apparent microbial growth – this makes me wonder if the biomass and C fluxes measured in the long-term incubations are comparable to those that would be measured in natural soils. I realize this is a study based on idealized conditions, but it is relevant to discuss similarities and differences compared to the real world.

Thank you for raising this point. We agree with the reviewer that this experiment is different compared to real soils. We modified the text to suggest caution when comparing our results with natural soils (lines 121-125 and lines 150-164). We think this study is an effort to evaluate the diversity – ecosystem function relationship in a soil-like matrix where the level of complexity was greater than in previous studies (more species than previous studies ~10-100). We are aware that more effort is needed to further bridge the gap between lab experiments and field conditions.

Other comments

L85: at this point in the manuscript, it is not clear how aggregation is connected to substrate supply – some explanations are needed

We have added a short explanation in line 87-88 and modified the results and discussion session to make it clearer (lines 180-190). We observed a positive relationship between soil aggregation and respiration, growth and CUE in dry soils (Supplementary Fig. 11), but not in wetter soils. Thus, we are suggesting that the microorganisms in dry soils were more resource limited and that is why we observed these positive relationships in dry but not wet soils.

L102: in the Methods (P13, L265), dilution is reported to be of four (not five) orders of magnitude

Sorry, we corrected this mistake. The dilution for D2 was 5. (line 300).

L138: does “scaled” mean “increased”?

Yes, we rephrased this sentence (line 137-140).

L153: perhaps “is consistent with” instead of “corroborates to”?

Thanks for making this suggestion. Corrected.

L157: this hypothesis is a bit far fetched and not testable here – especially the link to CUE – because diversity is manipulated but not complementarity. The hypothesis is plausible, but some more explanations of why CUE should increase when species are complementary are needed

Thanks for noticing this caveat in our explanation. We revised this explanation in lines 167-180. Two mechanisms are normally used to understand the biodiversity-ecosystem functioning relationship. The first is the “selection effect” which includes processes that at high diversity selects for species that have the capacity to perform more efficiently a function and contribute positively to the community productivity. The second process is the “complementarity effect”. This includes facilitation and niche differentiation that results from species interactions and increase community productivity in higher diversity levels. Because we found a positive influence of diversity only in wetter soils, and moisture can modulate inter-species interactions (de Vries et al., 2018 and Tecon et al., 2018), we are suggesting that complementarity mechanisms might be responsible for the positive influence of diversity on CUE. Our results do not indicate “selective effects” because no relationship was observed between diversity and CUE in the dry soils. Another relevant aspect to keep in mind is that there will only be a positive impact of diversity on CUE if growth increases faster than respiration. A previous study suggested that complementarity effects responded to changes in abiotic conditions (e.g. temperature) impacting positively growth in liquid cultures (Garcia et al., 2019).

References:

de Vries, F. T. et al. Soil bacterial networks are less stable under drought than fungal networks. *Nature Communications* 9 (2018)

Garcia, F. C., Bestion, E., Warfield, R. & Yvon-Durocher, G. Changes in temperature alter the relationship between biodiversity and ecosystem functioning. *Proceedings of the National Academy of Sciences* 115, 10989–10994 (2018)

Tecon, R., Ebrahimi, A., Kleyer, H., Levi, S. E. & Or, D. Cell-to-cell bacterial interactions promoted by drier conditions on soil surfaces. *Proceedings of the National Academy of Sciences* 115, 9791–9796 (2018)

L179: typo “effect on CUE”

Thank you!

L181: I do not see the connection between these sentences. The premise is that in the long-term it is difficult to separate abiotic vs. biotic drivers on CUE. As explained elsewhere in the manuscript, long-term and short-term experiments tell different parts of the story, but a short-

term experiment won't help disentangling the long-term abiotic vs. biotic effects on CUE

We have modified the text to improve clarity (lines 200-207). The short-term experiment was needed to evaluate the physiological response to changes in moisture and temperature without changes in the microbial community. While long-term changes will result in distinct communities, short-term changes will induce a physiological response.

L187: why selecting the dry D0 treatment?

From a feasibility point of view, we could perform the short-term experiment with only one treatment. We randomly decided on the DO dry and two temperatures. Selecting for one diversity treatment, one moisture level, two temperatures, and applying all different abiotic combinations resulted in 80 samples. The incubation for the ^{18}O -H $_2\text{O}$ CUE method is normally done over a 24 hr period and we could not run more samples in this short timeframe.

L247: or the other way around? From previous statements and structural equation modelling, it seems that abiotic factors are acting indirectly on microbial metabolism, the effects being mediated by microbial community characteristics – so it is the abiotic effects that are masked by the biotic ones

We performed structural equation modelling to infer causality between the variables. In our hypothesized model we have paths from the abiotic variables (temperature and moisture) going directly to CUE (Supplementary Fig. 14). If the abiotic variables would be directly driving CUE, these paths would be significant in our model, but they were not. This implies that the biotic variables are driving the communities and the community parameters are impacting CUE directly. We do agree with the reviewer that abiotic factors indirectly drive CUE and we did phrase it like that in the abstract and in the main text (lines 233-240). Because CUE is capturing microbial growth and respiration, it should ultimately be driven by the microorganisms.

L248: this work shows that microbial diversity matters, but why “more than previously believed”? this seems a bit of an overstatement...

We have modified the sentence.

L264: shouldn't the inoculated amount be expressed as a volume of inoculum solution per gram of model soil? Or was 'real' soil mixed with the model soil? Please clarify

We appreciate your suggestion. We rephrased the sentence (lines 290-294). The reviewer is right, we inoculated these model soils with an inoculum and not by mixing natural soils with the model soil. We decided to refer to the proportion of natural soil added to the model soil as an inoculum because we thought it is more informative to know how much natural soil was added to each microcosm, rather than saying that 2 ml of a soil suspension was added to each microcosms. We modified the sentence to improve clarity.

L275: higher WHC should correspond to higher (less negative) water potential

Thanks for noticing our mistake! We have corrected it (line 305)!

L305: I would add a short sentence to explain that in addition to the long-term incubations, you also want to perform short-term incubations to test abiotic effects on rates independent of community changes. It is stated later, but I found this transition not very clear

Thanks for the suggestion, we have added a short description there (lines 322-326).

L337: typo “establish”

Thank you.

L334: typo “capillarity”

Thank you.

L348: the sentence starting here is convoluted and hard to follow

We have added the formula to make it clearer and revised the text to improve clarity (lines 384-386).

L466: reference missing author list

We corrected the mistake.

Figure 1: please check dilution value – 5 or 4 orders of magnitude?

Thank you, we corrected the value in the text; it is 3 and 5 orders of magnitude for D1 and D2, respectively (lines 295-296).

Figure 2 (also some figures in SI): data points are clustered at low diversity levels, and variance does not seem homogeneous – how were these inhomogeneities in the data treated when conducting the regression analysis? Also, the CUE function in the bottom right panel seems to completely miss the high CUE values at high diversity, suggesting that the model chosen for the fitting is not appropriate

We added some extra information regarding our statistical analysis in the “Statistical Analysis section” (lines 402-409). We checked for variance homogeneity and when appropriate, the data were log transformed. Regarding the CUE model fit: the relationship between diversity and CUE was evaluated first by comparing the log and negative exponential curves, and then upon seeing that these underpredicted CUE at high diversity, we performed a break-point analysis for verification. The break-point analysis revealed that after phylogenetic diversity of 4.48, there was no relationship between CUE and diversity.

We started fitting a log and a negative exponential function to the data because mathematically CUE values cannot be higher than 100%, and some studies have shown that there is a biological maximum of ~80%. Thus, CUE values should saturate at some point; the question is at which level of diversity does this happen. To make this point more clearly, we have added a description of the break-point analysis. We also modified the description in the results and discussion to make this clearer (lines 141-145 and 154-166).

Figure 3: I would suggest highlighting the bars in the box plots that correspond to unchanged abiotic conditions in the long- and short-term incubations. These are reference points for the

other bars

We appreciate the suggestion. We modified the Figure 3 accordingly.

Reviewers' Comments:

Reviewer #1:

Remarks to the Author:

Domeignoz-Horta et al. have successfully addressed my concerns. I have no further comments on the manuscript. Congratulations!

Xin Jing

4/27/2020

Reviewer #2:

Remarks to the Author:

The authors have addressed my main initial comments and I find the manuscript improved. However, I still have some relatively minor suggestions/comments:

General: it seems that supplementary Figures are not referred to in the same order in which they are presented in supplementary information.

L44: several papers have studied microbial growth and respiration at different soil moisture levels, though not all have explicitly reported CUE-soil moisture relations (e.g., Anderson and Domsch, 1986; Göransson et al., 2013); I would not state that the impact of moisture changes on microbial metabolism is limited to two studies.

L45: osmoregulation is performed via synthesis or accumulation of osmolytes, not via EPS, which have other functions (maintaining hydration and solute transport).

L71: hypothesis 3 is rather vague and hard to falsify – could it be formulated in a more specific way, indicating an expected direction of change of CUE with varying environmental conditions?

L86: suggested amendment “fungal:bacterial ratio”

L87: connectivity relies on water-filled pores, so it can be higher within aggregates in dry conditions, but in wet conditions when large inter-aggregate pores are filled and water is moving I would argue that connectivity is highest among aggregates.

L135: it is useful to remind about the hypothesis, but then I would suggest doing it in the previous paragraph, where there is already a sentence on diversity effects on CUE (L130).

L142: I would include this threshold as a vertical line in Figure 2.

L224: how about osmolytes? They would also be missed with a DNA-based method if the relation between biomass and DNA was assessed at a time when osmolytes were not produced.

L253: but see more recent papers showing a different pattern (Soares and Rousk, 2019).

L276: the last sentence of the conclusion is rather generic and does not point to the main message of the paper – I would remove it.

L382: minor detail – use Greek symbol ‘ μ ’, not ‘u’ in micrometer

L385: the MWD definition should be amended – remove the equal sign (this is not an equation), and place “i” as subscripts.

L386: based on the formula above, “X” should have an overbar indicating averaging.

L407: note that the logarithmic function is monotonically increasing, but does not saturate; the negative exponential function is monotonically decreasing, so not a good choice for data showing and increasing pattern.

L408: are the break point calculated based on piecewise linear functions?

L427: which “bacterial data” are referred to here? Diversity or CUE data, or both?

References

Anderson, T.H., Domsch, K.H., 1986. Carbon assimilation and microbial activity in soil. Z. Pflanzenernahrung Bodenkd. 149, 457–468.

Göransson, H., Godbold, D.L., Jones, D.L., Rousk, J., 2013. Bacterial growth and respiration responses upon rewetting dry forest soils: Impact of drought-legacy. Soil Biol. Biochem. 57, 477–486. <https://doi.org/10.1016/j.soilbio.2012.08.031>

Soares, M., Rousk, J., 2019. Microbial growth and carbon use efficiency in soil: Links to fungal-bacterial dominance, SOC-quality and stoichiometry. *Soil Biol. Biochem.* 131, 195–205.
<https://doi.org/10.1016/j.soilbio.2019.01.010>

We are grateful to the reviewers reading and comments on our article. Their comments helped us to improve the article. Please find below our answer to every comment. As requested, we highlighted in the article file (in orange) the changes that were made in the text. When appropriate, we refer to the line numbers in the newly revised version of the manuscript.

REVIEWERS' COMMENTS:

Reviewer #1 (Remarks to the Author):

Domeignoz-Horta et al. have successfully addressed my concerns. I have no further comments on the manuscript. Congratulations!

Xin Jing
4/27/2020

Reviewer #2 (Remarks to the Author):

The authors have addressed my main initial comments and I find the manuscript improved. However, I still have some relatively minor suggestions/comments:

General: it seems that supplementary Figures are not referred to in the same order in which they are presented in supplementary information.

Thank you. We corrected this mistake.

L44: several papers have studied microbial growth and respiration at different soil moisture levels, though not all have explicitly reported CUE-soil moisture relations (e.g., Anderson and Domsch, 1986; Göransson et al., 2013); I would not state that the impact of moisture changes on microbial metabolism is limited to two studies.

We agree with the reviewer that various studies have evaluated the impact of moisture on CUE. In this paragraph we are focusing specifically on moisture and temperature effect on CUE. Thus, while various studies looked into the impact of abiotic factors on other components of microbial physiology, to our knowledge only two studies have investigated the impact of moisture on CUE. We modified the sentence to clarify that we are specifically referring to CUE (L47).

L45: osmoregulation is performed via synthesis or accumulation of osmolytes, not via EPS, which have other functions (maintaining hydration and solute transport).

We modified the sentence to refer to EPS as another response to drought. The modified sentence is: Normally, soil microbial communities living in drier soils are expected to have higher metabolic costs due to osmoregulatory mechanisms such as production of intracellular solutes (Harris 1981). Another response to drought is the production of extracellular polysaccharide (EPS), which might also imply in further costs (Manzoni et al., 2012a) (L47-50).

L71: hypothesis 3 is rather vague and hard to falsify – could it be formulated in a more specific way, indicating an expected direction of change of CUE with varying environmental conditions?

Thanks for noticing this caveat. We modified hypothesis 3 (L75).

L86: suggested amendment “fungal:bacterial ratio”

We modified the sentence according to reviewer’s suggestion (L89).

L87: connectivity relies on water-filled pores, so it can be higher within aggregates in dry conditions, but in wet conditions when large inter-aggregate pores are filled and water is moving I would argue that connectivity is highest among aggregates.

Thanks for noticing this caveat in our explanation. We agree with the reviewer point. We modified sentence reads: Soil aggregation is measured as a proxy for substrate supply. For example, under low water content connectivity is greater within than between aggregates while under higher water content connectivity is increased between aggregates than within an aggregate (L90).

L135: it is useful to remind about the hypothesis, but then I would suggest doing it in the previous paragraph, where there is already a sentence on diversity effects on CUE (L130).

We replaced the sentence remembering our hypothesis following the reviewers suggestion (L136).

L142: I would include this threshold as a vertical line in Figure 2.

Thanks for the suggestion. We modified the figure accordingly.

L224: how about osmolytes? They would also be missed with a DNA-based method if the relation between biomass and DNA was assessed at a time when osmolytes were not produced.

Thanks for noticing this caveat in our explanation. We added the production of intracellular solutes into this sentence. The new sentence is: A previous study showed that CUE decreased with drought duration (Tiemann, 2011), which could be associated with extra costs due to desiccation and the production of intracellular solutes or extracellular polysaccharides (EPS) both not captured by the DNA-based method of growth measurement used here (L228).

L253: but see more recent papers showing a different pattern (Soares and Rousk, 2019).

We modified the sentence to account for this more recent study. The sentence is now: We found fungal:bacterial ratio did not impact CUE in contrast to a previous study (Soares & Rousk 2019). This difference might be due to the lower fungal:bacterial ratio in our artificial soil compared to natural soils (Soares & Rousk 2019) (L256).

L276: the last sentence of the conclusion is rather generic and does not point to the main message of the paper – I would remove it.

Thanks for the suggestion. We deleted the last sentence.

L382: minor detail – use Greek symbol ‘mu’, not ‘u’ in micrometer

Thanks for noticing this mistake.

L385: the MWD definition should be amended – remove the equal sign (this is not an equation), and place “i” as subscripts.

Thanks for noticing this error! We corrected the formula.

L386: based on the formula above, “X” should have an overbar indicating averaging.

We added the overbar to indicate averaging (L384).

L407: note that the logarithmic function is monotonically increasing, but does not saturate; the negative exponential function is monotonically decreasing, so not a good choice for data showing and increasing pattern.

Thanks for asking this question. We decided not to show our data with the logarithmic function because – as you point out – it is monotonically increasing. We initially fit this equation with the hypothesis that the diversity-CUE relationship would not fully saturate over the range of observed diversities. As we saw, this is not the case. In the latter case of a negative exponential function, I think we miscommunicated. The exponent in the equation is negative ($y = m * \exp(-x) + b$). We have modified the sentence in the text to make it clearer that this is a “saturating exponential function” (L409).

L408: are the break point calculated based on piecewise linear functions?

Yes, the break point analysis was calculated using a piecewise regression approach with the Segmented package. We modified the text to refer to piecewise regression as it is probably a more common used term (L411). Thank you for asking!

Reference:

Muggeo V. M. R. 2003. Estimating regression models with unknown break-points. *Statist. Med.* **22**:3055-3071.

L427: which “bacterial data” are referred to here? Diversity or CUE data, or both?

We are referring to the bacterial diversity. The modified sentence reads: Because we observed a non-linear relationship between bacterial diversity and CUE, we log transformed the bacterial diversity data for the SEM (L430).

References

Anderson, T.H., Domsch, K.H., 1986. Carbon assimilation and microbial activity in soil. *Z. Pflanzenernahrung Bodenkd.* 149, 457–468.

Göransson, H., Godbold, D.L., Jones, D.L., Rousk, J., 2013. Bacterial growth and respiration responses upon rewetting dry forest soils: Impact of drought-legacy. *Soil Biol. Biochem.* 57, 477–486. <https://doi.org/10.1016/j.soilbio2012.08.031>

Soares, M., Rousk, J., 2019. Microbial growth and carbon use efficiency in soil: Links to fungal-bacterial dominance, SOC-quality and stoichiometry. *Soil Biol. Biochem.* 131, 195–

205. <https://doi.org/10.1016/j.soilbio.2019.01.010>